# Urban Itineraries with Smartphones to Promote an Improvement in Environmental Awareness among Secondary School Students

**DOI:** 10.3390/ijerph20032009

**Published:** 2023-01-21

**Authors:** Juan-Francisco Álvarez-Herrero

**Affiliations:** Department of General Didactics and Specific Didactics, University of Alicante, 03690 Sant Vicent del Raspeig, Spain; juanfran.alvarez@ua.es

**Keywords:** didactic itineraries, environmental education, urban science, sustainability, secondary school students, educational technology, ICT, smartphones

## Abstract

Our world is undergoing a series of changes that are taking it to an unsustainable situation. In addition to alerting the population, we must seek education towards a more sustainable world. This research proposes the realization in Alcoy (Spain) of some urban itineraries with mobile devices and with secondary school students, in order to promote an improvement in awareness and action on environmental problems. This activity aims, among other objectives, through informal learning and outside the classroom, to raise awareness among secondary school students about the environmental problems that threaten us. With the completion of a questionnaire, after carrying out these urban itineraries, the results of 214 students confirm that, with this type of activity, there is a notable improvement in the level of awareness and concern for environmental problems. It is also detected that men prefer these types of environmental awareness tests, compared to women, who prefer tests that have a more creative and artistic theme. It is worth continuing to propose this type of activity among students and improve their approach by forecasting, planning, and improving the training of the teachers involved in it.

## 1. Introduction

### 1.1. Environmental Education and Educational Technology

The awareness of the population around the problem of the continued deterioration of our planet and the need to contemplate a solution that leads to a more sustainable world is something urgent and everyone’s task. Despite the existence of sustainable development goals (SDGs), these, unfortunately, are often not known by society in general and, of course, are much less frequently put into practice or taken into consideration in daily actions [1]. It is precisely in schools and educational centers where more and better results can be achieved on the path to making our planet more sustainable [2,3,4]. In Spain, although a couple of decades ago, the existing educational legislation did recognize the importance of working on environmental education with students as a cross-cutting subject. For a few years now, environmental education has barely been considered in the study plans of the different pre-university and university studies [5]. We found educational centers, administrations, groups, and teachers who, in their centers, do carry out initiatives around working on environmental education and the SDGs in the classroom [6,7,8,9,10]. However, these initiatives tend to be sporadic, without continuity, and they are very scarce in all the centers as a whole. Educational administrations, both central and those of the various Spanish regions, try to encourage, without much success, the participation of centers in networks and programs of sustainable centers [11]. With this, it is intended that, in some way, awareness of the need to educate and take measures against this problem reach as many people as possible. However, there is still a lot of ignorance and little action in the educational world around the SDGs and environmental education [8], even when dealing with problems and actions that take place in the most immediate environment of the students, such as their neighborhood or their city.

It is precisely in these environments close to the students where, many times, we find the environmental problems most ignored by people. One of them is noise pollution, which is very present in the streets of most cities and goes unnoticed or taken for granted. This noise pollution entails numerous problems for our health [12], and an excellent way to make students aware of its presence and its effects can be to measure and map it in the city where they live [13,14,15,16]. This assessment of noise pollution allows people to be aware of its presence and how important it is to reduce it to improve their health.

The presence of parks and green areas in the city also contributes to reducing the negative impact that noise pollution has on people’s health [17]. Additionally, these green areas, in turn, encourage us to incorporate native plants into the urban environment that improve knowledge of them among the citizens of said city. Thus, the use of mobile applications that allow for the recognition of these plants can contribute to an improvement in the learning and study of the environment, contributing, in turn, to the environmental education of the population and its necessary awareness. Currently, there are many applications for smartphones that allow this recognition of plants through the use of augmented reality, and there are already some pedagogical experiences in which use has been made of these and other digital technology applications to contribute to the environmental and sustainable education of the citizens [18,19,20,21].

Today’s young people live oblivious to the problems that surround them. Different natural catastrophes, as well as all kinds of catastrophes caused by human beings, can occur in the world, but if these do not affect them directly, these young people feel that they do not go with them. However, they do live very intensely in everything related to the use of smartphones [22]. Currently, almost all of the young people who are in secondary education in Spain have a mobile device and are usually in more than one social network [23]. Thus, within the social networks that they use the most are mainly the networks Instagram and TikTok [24]. Even on Instagram, we find young people who have more than one account. However, what is no longer so massive is the use of these social networks in teaching [22,23,24]. There are very few cases in which teachers use these social networks to improve the motivation and learning of their students [25,26,27]. It is a shame that a resource that is so well-accepted and interesting to high school students is not used more frequently in high school classrooms. There are still many teachers who see more dangers than virtues in the use of mobile devices in the classroom. Among the main reasons that are argued are: they cause distraction and dispersion of knowledge; they can generate addiction problems in students and risks to their personal integrity, with cases of cyberbullying or bullying, sexting, viral challenges, etc.; and, they generate an attitude and a culture of minimum effort, since these devices tend to excessively facilitate actions and processes that could otherwise generate learning (for example: not knowing how to look in an analog dictionary, not knowing how to tell the time on an analog clock, writing with many misspellings or with abbreviated text that only those of his generation understand) [28,29]. On the other hand, there are teachers who find in the smartphone an excellent resource to motivate and improve the learning of their students and find multiple advantages that justify its necessary use in the classroom: it is a resource that is both a very large set of tools (compass, photo and video camera, clock, GPS, telephone, music and video player, encyclopedia, dictionary, notepad, language translator, agenda, calendar, etc.); it is very versatile and easy and logical to use; it enables ubiquitous learning; it has multiple advantages: immediacy, you can perform many more tasks and functions by installing apps and/or attaching sensors or accessories. All this leads many teachers to make pedagogical use of these devices, with results that provide improvements in student motivation and learning [30,31,32,33,34].

Combining the two topics discussed so far, environmental education and the use of mobile devices among secondary school youth, a symbiosis can be achieved in which learning and awareness of the planet’s sustainability benefit enormously, generating multiple advantages with this [35].

Álvarez and Vega already verified in 2010 that environmental education taken to high school students with constructivist and non-traditional activities manages in these students to generate awareness and interest in the sustainability problems of our planet [36]. However, the sustainability of the planet remains an issue of little concern, more specifically among high school students [37,38].

### 1.2. An Example of Activity to Promote Learning

An activity of these characteristics is, for example, the one that has been carried out in different cities in Spain, and even in the rest of the world, for almost a decade, and is the one that is part of the project: World Mobile City Project (WMCP) [39]. This project, born in 2013 in the hands of the Lacenet teachers’ association, aims to train students in the management and more profitable use of smartphones, allowing them to know how to get around a city, knowing its characteristics, how to get around with public transport, interact with their citizens, develop their digital competence, etc. [40,41,42]. In the city of Alcoy (Alicante, Spain), this activity has been carried out under the name of Alcoyanada and adhered to the WMCP project. In this activity, the participating students, normally students of the last years of Educación Secundaria Oligatoria (ESO) or of the two high school years, are organized in small groups of 5 or 6 students, and during one morning, they carry out an itinerary or route through the streets of Alcoy as a gymkhana. The objectives of this activity are: to improve the knowledge of the city of Alcoy, learning to move around the city through public transport, with the help of mobile devices; use mobile devices for purposes of social and educational interest; and make use of public transport, knowing its advantages and how to use it [43]. All this, as reflected in this previous investigation [43], entails a series of interesting advantages:Get to know the most immediate urban environment for students;Know the culture, art, history, and the urban natural environment;Identify and deepen knowledge of points of interest in the city of Alcoy;Learn to find your way around the city, both on foot and by public transport;Develop digital competence, both in the use of mobile devices, in augmented reality and geolocation applications, in social networks, etc.;Live, socialize, and interact with the citizens of Alcoy;Work collaboratively with group members;Relate and interact with students of the same age and from other educational centers.

In the last two editions of this activity, among the tests to be carried out at the different points of interest to which the students access, the carrying out of activities that aim to raise awareness and act on issues of environmental education and do so from the city that is the space where this activity takes place. Specifically, the tests that have been incorporated aim to:Identify the trees around them by using an augmented reality application installed on their mobiles;Recognize and measure the noise pollution present in the city through the use of a mobile application that allows you to measure the sound decibels that occur in a specific place.

In this way, it is possible to not only raise awareness, but also to involve and mobilize students around an environmental education that seems to be forgotten in these times, which is so necessary to achieve some SDGs (especially Goal 3: Good Health and Well-being; Goal 4: Quality Education; Goal 5: Gender Equality; Goal 11: Sustainable Cities and Communities; Goal 12: Responsible Consumption and Production; Goal 13: Climate Action; Goal 14: Life Below Water; and Goal 15: Life on Land) that seem increasingly ignored by society in general—they are so essential to keep them in mind, so as not to reach an irreversible situation with our planet.

These types of activities, tours, itineraries, or urban gymkhanas, despite their great potential as a pedagogical resource to improve learning, are quite underused among the educational community [44,45,46,47,48]. Additionally, as has been seen, in addition to making it possible to leave the classroom and approach the urban environment, it allows for learning, awareness, and action against causes, such as the degradation of our planet and in favor of its sustainability.

### 1.3. Objectives

This research focuses on verifying the degree of acceptance of these type of activities, which are focused on working on environmental education among secondary school students and whether students perceive that these activities lead to an improvement in their learning and a greater awareness of the existing ones environmental problems in our cities.

Thus, the objective of this research is:-Check the acceptance of activities to promote environmental awareness in an urban environment among secondary school students.-Additionally, as specific objectives, it is also sought:Check if there are differences between the interests of men and women around activities of an environmental nature;Check if this type of activity generates interest and motivation to learn and, more specifically, due to the environmental problems of our cities.

## 2. Materials and Methods

The research presented here is divided into three phases:Prior to the activity,The activity or gymkhana (team competition game in which the participants must overcome a series of tests that can be of ingenuity, ability, physical, or sports along a route),After the activity.

Although the activity (Phase 2) is the part in which the objective of this research is put into play, to understand it, it is necessary to contemplate the other two phases.

Thus, in the phase prior to the gymkhana, a training course was held for the teachers who were going to participate in the activity. This training was sponsored by the Center for Training, Innovation, and Educational Resources (CEFIRE) of Alcoy, with a duration of 10 h and with the participation of 28 teachers from a total of 13 secondary schools in Alcoy and other nearby towns, such as Ibi, Muro de Alcoy, Bañeres de Mariola, Biar, Ontinyent, and Castalla. The teachers, although there were practically all specialties, were mostly teachers in the areas of science and technology. The age of the teachers was also very varied, from 28 to 58 years. Their participation in this training was completely voluntary, and they found out about it thanks to the dissemination that the CEFIRE makes of its training actions. In this training, in addition to addressing the technical and management issues necessary to subsequently carry out the gymkhana, strategies and dynamics were discussed with which to work and make students aware of the environmental problems of the city. This course was focused on dealing with the issues of: noise pollution, the preservation of diversity and more specifically of urban flora, uncontrolled dumping of urban solid waste, and other issues on the sustainability of the urban environment. In this way, it was ensured that these teachers would subsequently deal with both the technical and competence issues in digital technology and the environmental dynamics with their respective students. Strategies and dynamics were discussed with which to work and make students aware of the city’s environmental problems.

Additionally, the phase after the completion of the gymkhana had its focus on the evaluation of the activity. For this purpose, an ad hoc questionnaire was prepared and provided to all the participating teachers, so that they could distribute it among the students who participated in the activity.

### 2.1. Activity

In the days prior to the activity, the teaching staff of the participating centers, and after having been trained in the aforementioned training course, instructed their students in the different applications to be used, in the technical and functional requirements of mobile devices, and in all travel, organizational, and other issues, so that the students could get around the city autonomously on the day of the activity.

The activity takes place during a morning of a school day. The students that day do not have the usual classes and dedicate the whole morning to this activity. Since it is carried out outside the educational centers (in an urban environment: the city of Alcoy), it is considered an extracurricular activity. However, it is an activity that will be evaluated (through rubrics carried out in the previous training) and taken into account by the participating teachers in their respective subjects.

In this, helped by their smartphones, as well as by public transport and making use of augmented reality and social networks, they have to complete a route through 6 different points of the city. At each of these points, they must perform a test and also post a photo or a short video on the social network Instagram [29]. Said photos and videos must be labeled with the hashtags of the activity: #ALCada22 and #WMCP22 (in the case of this latest edition). Participation in the latest editions, since this gymkhana is held every year, has been quite numerous and has had an average of 9–10 educational centers from Alcoy and other nearby cities (Bañeres de Mariola, Muro de Alcoy, Castalla, Biar, Ibi), as well as a total of 400–500 students participating in each edition.

In the gymkhana, secondary school students, organized in small groups (5–6 students), had to move around the city of Alcoy, with the aim of completing an itinerary with 6 points of interest that were assigned to them by means of cards (6 in total, 1 for each point of interest to visit, including point geolocation data, one question, and a related clue, as shown in Figure 1). The cards had been distributed at the beginning of the day. Between 9:00 a.m. and 1:00 p.m. of the same day they had to complete this route.

There are a total of 60 different points of interest, each with its own card, which, as they are printed in several sets of 60, allow them to be mixed and randomly distributed 6-by-6 to the groups of participating students, thus generating many possible combinations of different paths. In addition, each group of students had a transport card that allowed them to make, free-of-charge, as many trips as needed by all the people in the group with urban buses and during that day. In order to monitor and control the movements made by the students during the activity, they were monitored at all times by their teachers, since they had an application (Life 360) installed on their mobile devices that allowed them to know their location in real time. Likewise, each participating student was provided with an identification tag for the activity, and the relevant permits had also been requested from both the local police and the national police and civil guard of Alcoy. The students also had the necessary applications installed on their smartphones to carry out a series of tests that they had to carry out, one test at each point of interest, choosing one of the following (see Table 1) at each point:

The three videos and the three photos resulting from their passage through the 6 points of interest on their route had to be uploaded from open accounts to the social network Instagram and with the labels or hashtags of the day, #ALCada22 and #WMCP22, generating between all participating students a collaborative and geolocated map with images and videos of the entire city of Alcoy, represented by the 60 selected points of interest. The activity in the streets of Alcoy had to end at 1:00 p.m., since at that time, all the participants had to go to a sports center in the center of the city, where once everyone was present, a small closing ceremony was held with the delivery of a small detail to the participants.

As has already been mentioned, once the activity was finished, the participating teachers were asked to invite their students to fill out a short questionnaire to evaluate the activity in the two days following the gymkhana.

### 2.2. Data Collection Instruments

Ad hoc questionnaire was prepared through Google Forms (as can be seen in Appendix C), including three socio-demographic questions (sex, age, and school), and 8 students’ perception questions in a five-point Likert scale about: the whole experience (general objective of this research); each of the 6 activities in the 6 points of interest visited (specific objective 2); students’ self-perception regarding improvement of their awareness and concern for environmental problems (specific objective 1). Finally, the questionnaire included three open questions: one for the overall qualitative assessment of the activity by the students and another two in which they had to highlight those aspects that they liked the most and least. The quantitative results were treated with the statistical software IBM SPSS version 25 and the qualitative ones were treated with the ATLAS.ti 9 program.

### 2.3. Participants

As already indicated, prior to the gymkhana, a training course was held with 28 teachers from 13 educational centers, although in the end, 4 of these teachers, belonging to 4 other centers, did not participate in the gymkhana with their students, making the total number of participating teachers in the end 24 and the number of centers 9.

The participating students were from a total of 331 belonging to 9 secondary schools. Seven of these centers were from the city of Alcoy itself (CE01, CE02, CE03, CE04, CE05, CE06, and CE07), and the other 2 from the nearby cities of Bañeres de Mariola and Muro de Alcoy (CE08 and CE09), as can be seen in Table 2.

The vast majority of the students participating in the gymkhana had barely heard of the preparation of this SDG activity before. Additionally, the use they made of their smartphones was very limited to the use of WhatsApp and social networks, practically all of them unaware of other uses that can be given to the mobile, such as geolocation, augmented reality, the use of applications to know the position of the urban buses in real time, etc. The previous experience in the use of smartphones for educational use was practically nil. Additionally, they did not know about the trees and plants in their city and had not heard of noise pollution before.

In the evaluation questionnaire, of the total number of students participating in the gymkhana (*n* = 331), given its voluntary nature, 214 participated (*n* = 214), 64.6%, which makes it a significant sample, with a level of confidence of 99% and a margin of error of 5.28%. Of the 214 students who participated in the post-activity evaluation questionnaire, it was found that there was a greater presence of women (*n* = 110, 51.4%), compared to men (*n* = 96, 44.9%), and a small percentage of non-binary sex, 3.7% (*n* = 8). Possibly, this last number is not so high and is due to the fact that the students have answered in a non-serious way. The age of the participants ranges between 14 and 17 years (average 15.34 years, mode 15).

Table 3 compares the total number of students who participated with those students who completed the activity evaluation questionnaire.

The participation of the centers in the evaluation questionnaire was not homogeneous, and in three of them (C01, C07, and C08), it was below 50% and it also coincided that they are the same 3 centers that had a participation below the average.

## 3. Results

### 3.1. Overall Results

In the global evaluation question of the activity as a whole, using a Likert-type evaluation scale that goes from 1: not at all to 5: a lot, the #ALCada22 activity obtained an average value from the students of 3.35 (deviation typical: 1.147 and variance: 1.316). For the question that collected the degree of improvement in their level of awareness and concern for environmental problems, they obtained a much higher average than that of the activity as a whole (slightly more than 1 point above), with this being 4.37 (standard deviation: 0.573 and variance: 0.328). The average assessment of each of the activities proposed in the gymkhana can be seen in Table 4, with their corresponding codes, where F01, F02, and F03 correspond to the tests with photos and V01, V02, and V03 to the tests with videos.

From these latest results, it can be deduced that the activity, as a whole, enjoys an average rating higher than those obtained by each of the tests carried out individually. Additionally, within these tests, the two that had the greatest acceptance were the making of a video with some typical dance of Moros and Cristianos, as well as the making of a BookFace. The two tests that are closely related to environmental education and urban environmental sustainability, although they were not the worst valued, were placed in a discreet third and fifth place.

### 3.2. Results According to the Different Variables

It is interesting to see the results obtained when contemplating the variations that exist in the assessment of both the activity as a whole, and each of the tests were carried out individually. Thus, in Table 5, we can see this variation according to the sex of the participants and in Table 6 according to age.

As can be seen in the results of Table 6, in relation to the low scores obtained in all the categories for the non-binary sex, and as previously mentioned, the answers given have very possibly been made by students who have taken the evaluation questionnaire with little seriousness and have responded in a toxic way, so we are not going to take them into account.

Regarding the differences found between men and women, it is quite notorious to highlight that, in regard to the two tests that interest us on environmental education and for sustainability, it can be observed that, in the case of women, they find a value that presents 37 and 42 tenths less than in the case of men. For the tests of a more artistic and creative nature (such as the BookFace or the recreation videos of typical party dances), it is in the women where scores are obtained that are 10 and 38 tenths higher than those of the men. Additionally, notable, in a negative sense, is the ever-recurring worse assessment, both in men and women, of the test of recitation of poems in the native language. The Student’s *t*-test has been carried out for all the activity tests, and it is, indeed, verified that the differences are significant between men and women both in the tests in question: noise pollution record (t = 2.049; *p* = 0.042) and in the plants identification (t = 2.476; *p* = 0.014), such as the video recording tests of a typical dance (t = −2.329; *p* = 0.021) and of a video interview (t = −2.178; *p* = 0.031).

In the videos, although it is confirmed in Table 6 that, between 14 and 16 years of age, the older the participant, the worse the assessment obtained from said tests; in the case of proofs with photos, one cannot speak of any regularity.

### 3.3. Qualitative Results

Of the qualitative results, special attention must be paid to those related to the purpose of this research and the tests that had to do with the environmental and sustainable education of the students.

The responses were categorized into different categories, according to their thematic. In general terms, the general evaluation of the activity was clouded by many comments that highlighted the great inconvenience represented by the presence of rain throughout the activity (*n* = 121) and that, at times, it was very annoying. On the positive side, the development of the digital competence of the students (*n* = 148) and the carrying out of activities outside the center (*n* = 135) stood out, for the most part.

Going on to assess those positive aspects indicated by the students, the most outstanding and frequent ones, concerning the topic at hand, were:Live outside the classroom with other students (*n* = 76),Learn interesting things about my city (*n* = 53),Learn to use public transport (*n* = 41),Know and identify the trees and plants of my city (*n* = 32),Identify those areas of my city with the highest noise pollution (*n* = 27).

Some examples of these comments were:
“*What I liked the most was being able to learn in a fun way, outside the center, to identify plants and trees in my city, as well as to measure the levels of noise pollution that we find in the streets.*”—ST061
“*I really liked the tests, especially those in which we had to use mobile applications to identify trees and measure the noise of cars and motorcycles.*”—ST124

## 4. Discussion

Although the majority of studies carried out on the use of didactic itineraries as a resource for learning coincide in affirming that the use of smartphones, augmented reality, geolocation, and social networks help to improve interest, motivation, and learning among students [35,46,49,50,51,52,53]; on this occasion, we cannot affirm the same, since a very discreet result was obtained. One of the reasons that explains this poor result, compared to a notable result obtained on a previous occasion with a very similar approach [35], is the fact that, on the day of the activity, despite having been postponed and rescheduled twice for the same reason, the rain made its appearance, with this being, at some times of the day, quite intense and annoying.

However, it can be celebrated that the assessment that students give to the degree of improvement in their awareness and involvement with the environment and sustainability in an urban environment reached a remarkable value. In similar actions, with secondary school students and the use of mobile phones, changes in opinion and awareness about the socio-environmental challenge were also observed [54]. Additionally, it can be celebrated that the tests related to this aspect are well-valued, without becoming the most widely accepted, but not the least, either. It is worth noting that, although no notable differences were observed, with respect to the age of the participants, in the case of gender, it was observed that women value tests related to environmental issues worse than men; on the other hand, they value better than men those tests that are more related to creativity and artistic expression.

Another aspect that does not coincide with the studies carried out to date is that it seems that the preference for videos, over taking photos, has been taking place among young people in recent years [35,55,56,57], which already in this study is not so clear; thus, although there are some tests in which this is confirmed, the same does not happen with the BookFace test (F03), which triumphs over others that involve the making of a video.

It is clear that the use of digital technology and, more specifically, the use of smartphones with augmented reality, geolocation, and social networks allows us to motivate and generate interest between learning and a respect, appreciation, and awareness of a necessary action with the environment. This is made clear by other previous studies [35,58,59], and here, the more than remarkable assessment about the degree of improvement in their level of awareness and concern for environmental problems demonstrates and reaffirms it.

## 5. Conclusions

The use of gymkhanas or itineraries with mobile devices as a resource to take learning to the street, to the context and the reality closest to the students, that is the city, has always given positive results in all the experiences carried out. Additionally, having a consistent and combined use of digital technologies brings great benefits when it comes to secondary school students, since they live immersed in a world where they are heavily involved in the use of smartphones and social networks. If, in addition to all this, it is intended to improve awareness and appreciation of the environment, making students aware of the problems with which we find ourselves and promoting the production of an action, in addition to awareness, then the results are even more positive.

Activities such as the one carried out here show that urban sustainability is a possible solution close to adolescent students, which, although with evaluations that are not excellent, do allow us to predict that this is the way to train citizens who are prepared to promote a better world, i.e., more sustainable and environmentally friendly. Quite possibly, the non-appearance of the rain, as well as a better preparation of the participating teachers in environmental issues to be dealt with in the activity and in how to propose activities with their students based on the development of participatory and playful skills and strategies, would have allowed for better results.

For future interventions, both the training and planning on the dates to carry out the activity could be improved. Taking learning out of the classroom and making use of the urban environment is something that must continue to be performed and promoted in secondary schools and institutes, in order to allow students to get closer to the reality we live in first-hand and to know how to see those risks and problems that we face today, in order to tackle them in the present and the immediate future.

In future lines of research, it is intended to present the activity with numerous changes, which allow for an improvement in the assessment of the activity, as this will result in greater learning and greater environmental awareness.

## Figures and Tables

**Figure 1 ijerph-20-02009-f001:**
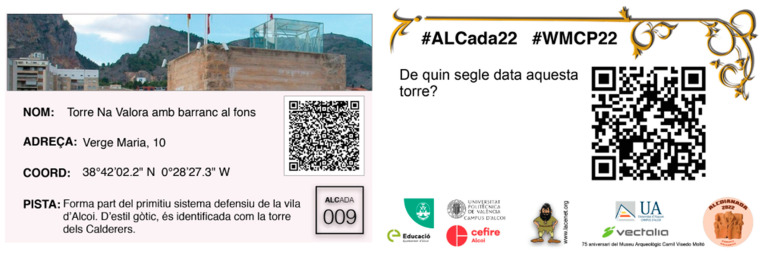
Example of the front and back of a card of one of the points of interest. Notes: Name: Na Valora tower with ravine in the background; Address: 10 Verge Maria street; Coordinates: 38°42′02.2″ N—0°28′27.3″ W; Clue: It is part of the primitive defensive system of the town of Alcoy. In Gothic style, it is identified as the tower of the boilermakers; #ALCada22 #WMCP22; What century does this tower date from?

**Table 1 ijerph-20-02009-t001:** Tests to be carried out by the students in the 6 points of interest.

Test	
F01	Take a screenshot with the highest record you get with the app: *Sound meter* at the point of interest you had to visit. Upload it as an image to Instagram and write in the comment the exact geolocation of the photo, putting the name of the place with the noise pollution record obtained. (Examples of this test can be seen in the Appendix A).
F02	Take a group photo with a nearby plant and upload it to Instagram, writing the common and scientific name of the species, using the app: *PlantNet* that allows you to identify trees, plants, or shrubs present at a point close to the point of interest that you had to visit (optional label: #ciutatverda). (Examples of this test can be seen in the Appendix B).
F03	Take a BookFace photo (You must take a photo integrating a book, body, and environment and upload it to Instagram). Try to be original. (optional tag: #BookFace)
V01	Do a dance, choreography, dance of Moros and Cristianos, etc. With or without costumes, with or without music. Make use of your imagination.
V02	Do a short video-interview with someone who passes by the point of interest you are visiting in which you ask that person: what was school like when they were a student?
V03	Make a video in which you are reciting verses in Valencian, by a well-known author or created by you.

**Table 2 ijerph-20-02009-t002:** Centers, teachers, and students participating in the activity.

Center	Activity	
	Students	Teachers
C01	35	3
C02	52	5
C03	35	3
C04	45	3
C05	23	1
C06	32	2
C07	30	2
C08	35	3
C09	24	2
Total	311	24

**Table 3 ijerph-20-02009-t003:** Students participating in the activity and the evaluation questionnaire.

Center	Activity		Questionnaire		Percentage of Participation in the Evaluation Questionnaire
	Frequency	Percentage	Frequency	Percentage	
C01	35	11.3	17	8.0	48.6
C02	52	16.7	45	21.0	86.5
C03	35	11.3	27	12.6	77.1
C04	45	14.4	40	18.7	88.9
C05	23	7.4	16	7.5	69.5
C06	32	10.3	26	12.1	81.2
C07	30	9.6	10	4.7	33.3
C08	35	11.3	9	4.2	25.7
C09	24	7.7	24	11.2	100
Total	311	100	214	100	68.8

**Table 4 ijerph-20-02009-t004:** Assessment of the tests carried out in the gymkhana.

	F01	F02	F03	V01	V02	V03
Average	3.12	3.06	3.25	3.26	3.12	2.78
Standard deviation	1.271	1.297	1.256	1.258	1.286	1.266
Variance	1.615	1.682	1.577	1.582	1.653	1.602

**Table 5 ijerph-20-02009-t005:** Evaluation of the activity and the tests carried out according to gender.

	Activity	F01	F02	F03	V01	V02	V03
Men	3.39	3.35	3.33	3.25	3.10	3.00	2.87
Women	3.43	2.98	2.91	3.35	3.48	3.34	2.77
Non-binary	1.75	2.13	1.88	1.88	2.00	1.50	1.63
Global	3.35	3.12	3.06	3.25	3.26	3.12	2.78

**Table 6 ijerph-20-02009-t006:** Evaluation of the activity and the tests carried out according to age.

Years	Activity	F01	F02	F03	V01	V02	V03
14	3.70	3.27	3.32	3.54	3.68	3.59	3.00
15	3.51	3.44	3.44	3.60	3.35	3.20	2.82
16	2.99	2.70	2.60	2.76	2.88	2.75	2.48
17	3.19	2.81	2.48	2.81	3.33	3.10	3.14
Global	3.35	3.12	3.06	3.25	3.26	3.12	2.78

## Data Availability

The data presented in this study are available on request from the corresponding author. The data are not publicly available due to involve humans.

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
