# Peer review of "Urban Itineraries with Smartphones to Promote an Improvement in Environmental Awareness among Secondary School Students"

_ijerph, 2023, doi:10.3390/ijerph20032009_

Round 1

Reviewer 1 Report

Congratulations for a well-deserved research. That said, for this same reason, we strongly encourage the author (blinded peer review?!) to delve much more into the relationships between the state of the art review, objectives and hypothesis with the methodology employed and its restrictions. Also, we highly suggest the author to deepen into the analysis, results, discussions, further research and its limitations as well as its conclusions. Even more, give the complexity of the research proposed and the constraints experienced, these requirements are much more needed.

Author Response

We want to thank the reviewer for his work, because with his suggestions and comments he contributes to improving the quality of our work. Many thanks.

All sections of the article have been reworked. And as the reviewer suggests, the introduction (state of the art) and the objectives have been more profusely related to the methodology. Likewise, the analysis, results and discussions have also been deepened, to also deepen the conclusions taking into account the limitations that this research presents.

Reviewer 2 Report

This study proposes some educational activities with mobile devices, to promote secondary school students’ awareness and action on environmental problems. It is an interesting subject, but the paper needs improvements in order to become more scientifically sound; more specifically, its limitations regard the methodology (sample and research instrument description), the lack of specific research objectives, and the paper organization.

The term “digital technology” is very broad for the title; it could be replaced by mobile/smart-phones, the actual technology used in this study. The abstract needs to briefly mention the activity, the context (informal learning?) and the country.

Line 32, with “pre-university” studies does the author mean studies with students aged <18 years old? (compulsory education). In introduction, clarify the educational level (i.e., secondary education), since “students” is also used for elementary school and university students.

Some re-organization of the (sub)sections is needed.

The first section is long and includes a bit of everything. I suggest this to be split into two separate sections: (i) introduction: at the end of this section, the purpose of this study should be clearly stated/mentioned; and (ii) Background to the study (or literature review): including background international and national research studies (the literature review could be enhanced with some evidence on secondary school students’ awareness/opinions on similar environmental issues).

A large part of the introduction (Lines 96-145) describes an activity (its objectives, advantages, aim of the tests, etc.); this does not have a place in the introduction, it is better to be discussed under materials and method section.

The specific research objectives/questions (RQs) and the limitations of the study are missing. The author mentions the objective (purpose) of this research in lines 146-150, but it is somewhat vague. I suggest (i) the purpose of the study to be clearly re-written in a smaller sentence (and placed, as indicated above, at the end of the introduction), and (ii) two specific research objectives to be created.

Was the activity-intervention and consequently, the mobile phones’ utilization, carried out in informal or formal context (within which school subjects)? Were student responses linked to any type of school assessment?

The sample description (section 2.2) should be enhanced and presented before the tests (Table 1). Although the focus is on the activity (2nd phase - the intervention), phase 1 is also important; because teacher training is linked to the educational goals of the activity and the way they worked with their students. Thus, I suggest to expand on the description of the 24 participant teachers; e.g., what was their specialization, their age-range, did they participate voluntarily in this training, how were they recruited? Additionally, with regard to the participant students (N=214): did they have prior experience in using their mobile phones for educational purposes (or was it only for the purpose of this study/intervention?), what was their prior knowledge of the environmental issues tackled in the activity (since their age-range was 14-17). I strongly suggest the creation of a Table indicating the characteristics of the sample, including educational centers, teachers, and students.

The research instrument is useful to be shown in Appendix, to facilitate the readership of the paper. Who constructed the instrument questions? How these questions are linked to the goals of the activity tests?

With regard to the qualitative analysis, a brief indication is useful to be provided: e.g., was it content analysis with themes/categories? A Table indicating students’ major responses would add value to the study. Since these are very briefly mentioned, are these issues to be analyzed-presented in another paper?

Could this study have implications beyond the specific context?

There is not always clear distinction between the author’s claims and other researchers’ work. For example, in line 69, whose opinion is that? Are the advantages of the activity, as discussed in lines 111-121, the author’s opinion, or were these measured in some way?

With regard to English language, in some sentences the meaning is not clear; e.g., in abstract, the sentence “Likewise,…artistic” (lines 13-15) needs to be re-phrased, and “on this occasion we cannot affirm the same, since a discreet result is obtained that slightly exceeds the approved one” (lines 347-348). Minor points: line 62, “they” should be “them”; explain the term “gymkhana” (line 153); there are full stops in the middle of some sentences (e.g., in lines 149, 159, 386), while some words begin with capital letter after the comma (e.g., in line 215).

With regard to the use of mobile phones in secondary education context, I would suggest the author to also consult the following studies:

Väljataga, T.; Mettis, K. Secondary Education Students’ Knowledge Gain and Scaffolding Needs in Mobile Outdoor Learning Settings. Sustainability 2022, 14, 7031.

Nikolopoulou, K. Secondary education teachers’ perceptions of mobile phone and tablet use in classrooms: benefits, constraints and concerns. Journal of Computers in Education 2020, 7(2), 257-275.

Author Response

We want to thank the reviewer for his work, because with his suggestions and comments he contributes to improving the quality of our work. Many thanks.

All those modifications that have been made and in which all the proposals and suggestions made by the reviewer have been addressed are detailed below. In this way, an attempt has been made to correct those deficiencies in terms of methodology (description of the sample and the research instrument), the lack of specific research objectives and the organization of the article.

Although the activity makes use of more digital technology (QR codes, geolocation, social networks, image and video editing software, etc.) than just the use of smartphones, the title has been changed at the reviewer's suggestion.

The abstract has been completely reformulated so that it contains both the activity, its purpose, the context and the country in which it is carried out.

Initially, the introduction talks about any type of student (both primary, secondary and university), and later, when the subject of the research is already focused on secondary school students, this student body is talked about exclusively. This has also been qualified in the text so that it is clear when talking specifically about secondary school students.

The entire Introduction section has been modified in response to the suggestions made by the reviewer.

It has been structured in 3 parts. An initial in which a general introduction of the theme is made. A second one in which we talk about activities of the type that is going to be investigated, indicating the results that have been obtained with them. And a third in which the general objective and the specific objectives that are intended to be achieved with this research are discussed. Similar references and research have been added, as well as others capturing the views of high school students on similar environmental issues, as suggested by the reviewer.

In the methodology section, all the changes and suggestions made by the reviewer have also been addressed:

- The context of the activity has been described, in which subjects it was carried out, and the evaluation that was carried out from these by the teachers

- the first phase, the teacher training course, has been better described

- Students' prior knowledge on sustainability issues and the use of smartphones has been described

- A table has been built to describe the sample with the participating schools, students and teachers

- The instrument used for data collection has been included as an appendix, providing a better description of this instrument and its relationship with the research objectives.

- The section on results and qualitative analysis of the research has been redrafted, indicating the frequency of the most numerous responses concerning the subject of this research.

New paragraphs have also been drafted in both the Discussion and the Conclusions.

Those expressions in which the reviewer did not appreciate a distinction between the author's statements and the works of other researchers have been qualified, argued and redrafted. For example, lines 69-70; 111-121.

The entire document has been reviewed to detect and correct:

- sentences whose meaning was not clear (such as those suggested by the reviewer - lines 13-15; 347-348)

- change a "they" to a "them" (62)

- change some capital letters behind ","

- remove some "." in the middle of sentences

- and other typos detected.

Also, the term gymkhana has been explained (line 153)

At the reviewer's suggestion, the reference Valjataga - Mettis has been added

Reviewer 3 Report

The aim of this study was to verify the degree of acceptance that the activities focused on working on environmental education has among secondary school students and whether the students perceive that these activities cause an improvement in their learning and greater awareness. on the existing environmental problems in our cities.

The results of 214 students confirm that with this type of activity there is a notable improvement in their level of awareness and concern for environmental problems. Likewise, a small difference is detected that makes the type of tests used to promote this awareness, are better valued by men compared to women, who prefer tests with other themes and more creative and artistic. The author suggests continuing to propose this type of activity among students, and improve their approach from forecasting, planning and improving the training of the teachers involved in it.

The paper requires some editorial/technical amendments and linguistic corrections.

The literature presented in the references is relevant and up-to-date but there are some editorial errors or omissions. Namely:

– In case of the source no. 5, the authors of the paper should be ‘Márquez Delgado, D.L.; Hernández Santoyo, A.; Márquez Delgado, L.H.; Casas Vilardell, M..

– In case of the source no. 6, the title of the journal should be: ‘Profesorado: Revista de Currículum y Formación del Profesorado’.

– In case of the source no. 7, the authors of the paper should be ‘Mora Zapater, J.L.’.

– In case of the source no. 9, the authors of the paper should be ‘Aguirregabiria Barturen, F.J.; García Olalla, A.M.’.

– In case of the source no. 10, the authors of the paper should be ‘Benayas del Álamo, J.; Marcén Albero, C.’.

– In case of the source no. 11, the author of the paper should be ‘Montero Caro, M.D.’.

– In case of the source no. 11, the issue number should be: 23, instead of 24.

– The sources no. 8 and 12 include exactly the same publication.

– In case of the source no. 15, the data about the publisher is missing. It should be: Association for Computing Machinery.

– In case of the source no. 18, the complete page range should be: 157-65.

– In case of the source no. 18, the doi number is: https://doi.org/10.4103/1463-1741.134916.

– In case of the source no. 28, the doi number is: https://doi.org/10.1016/j.ijinfomgt.2018.10.001.

– In case of the source no. 31, there is the unnecessary closing bracket.

– In case of the source no. 37, the authors of the paper should be ‘Barlam Aspachs, R.; Ribas Soler, M.; Foixenc Pérez, N.; Rochera Villach, M.J.’.

– In case of the source no. 38, the doi number is: http://hdl.handle.net/11162/101571.

– In case of the source no. 42, the authors of the paper should be ‘López de Haro, F., Segura Serrano, J.A.’.

– In case of the source no. 44, the authors of the paper should be ‘Montero Pozo, J.; Óscar Jerez, G.’. And the editors should be: ‘Salido López, J.V., Salido López, P.V.’.

– In case of the source no. 45, the authors of the paper should be ‘Torralba Burrial, A.; Herrero Vázquez, M.’.

– In case of the source no. 45, the fragment of the title of the book should be ‘Edunovatic 2016’.

– In case of the source no. 47, the authors of the paper should be ‘Alcántara Manzanares, J.; Medina Quintana, S.’.

– In case of the source no. 50, the author of the paper should be ‘Martín Hernanz, S.’.

– In case of the source no. 50, the doi number is: http://hdl.handle.net/10486/671862.

– I am not expert in Spanish language but it seems that in case of the source no. 53, the authors’ names should be written without hyphen.

– In case of the source no. 54, the correct doi number is rather: https://doi.org/10.25267/Rev_Eureka_ensen_divulg_cienc.2018.v15.i2.2105.

– In case of the source no. 55, the correct doi number is: https://doi.org/10.29333/ejmste/100639.

– In case of the source no. 56, the authors of the paper should be ‘Ayerbe López, J.; Perales Palacios, F.J.’.

The results are presented in a clear way. The discussion and conclusion organized correctly.

Author Response

We want to thank the reviewer for his work, because with his suggestions and comments he contributes to improving the quality of our work. Many thanks.

Thank you for your initial comments.

All proposed modifications in the article's references have been made

By removing one of the duplicate [8] and [12] , as well as introducing one more reference proposed by Reviewer 2, they have all been now renumbered.

In the change proposed in reference 44 (now 43), the reviewer commented that one of the authors should be changed as: Óscar Jerez, G., . It has been changed, but it has been changed to Jerez García, O. which is really the name of the author.

About the reference 53 (now 53), regarding writing the names of the authors without a hyphen, we have not made this change since the hyphens are necessary, since what these authors do with these hyphens is to be able to group the two surnames that are found in Spanish culture, and thus you can easily differentiate from other authors who have the same first name. The initials of the names, if they have two names, are put without hyphens.

Insist on our gratitude to the reviewer and his commendable work.

Round 2

Reviewer 2 Report

The manuscript has now been improved and I consider it is ok for publication

Author Response

Dear reviewer

Thank you very much for your contribution to the improvement of my contribution.
As you have seen, I have taken into account all your suggestions and comments.

Many thanks

Best regards